# Analysis of the indispensable RAD51 cofactor BRCA2 in *Naganishia liquefaciens*, a Basidiomycota yeast

Maierdan Palihati, Hiroshi Iwasaki, Hideo Tsubouchi

**The *BRCA2* tumor suppressor plays a critical role in homologous recombination by regulating RAD51, the eukaryotic homologous recombinase. We identified the BRCA2 homolog in a Basidiomycota yeast, *Naganishia liquefaciens*. BRCA2 homologs are found in many Basidiomycota species but not in Ascomycota species. *Naganishia* BRCA2 (Brh2, for *BRCA2 h*omolog) is about one-third the size of human BRCA2. Brh2 carries three potential BRC repeats with two oligonucleotide/oligosaccharide-binding domains. The homolog of DSS1, a small acidic protein serving as an essential partner of BRCA2 was also identified. The yeast two-hybrid assay shows the interaction of Brh2 with both Rad51 and Dss1. Unlike human BRCA2, Brh2 is not required for normal cell growth, whereas loss of Dss1 results in slow growth. The loss of Brh2 caused pronounced sensitivity to UV and ionizing radiation, and their HR ability, as assayed by gene-targeting efficiency, is compromised. These phenotypes are indistinguishable from those of the *rad51* mutant, and the *rad51 brh2* double mutant. *Naganishia* Brh2 is likely the BRCA2 ortholog that functions as an indispensable auxiliary factor for Rad51.**

## Introduction

Mutations in the *BRCA2* gene are associated with a high risk of developing breast, ovarian, and other types of cancers (Wooster et al, 1995; Ozçelik et al, 1997; Breast Cancer Linkage Consortium, 1999; Antoniou et al, 2003; Edwards et al, 2003; King et al, 2003; van Asperen et al, 2005). Cells deficient in BRCA2 exhibit a spectrum of traits typically associated with defects in DNA repair, including hypersensitivity to genotoxic agents, gross chromosomal rearrangements, and frequent occurrence of chromatid breaks (Sharan et al, 1997; Yu et al, 2000). BRCA2 plays a critical role in homologous recombination (HR) by regulating RAD51, the eukaryotic RecA-like recombinase (Roy et al, 2011; Prakash et al, 2015). HR not only provides the error-free pathway for repairing DNA double-stranded breaks (DSBs) but also plays an important role in protecting replication forks (Schlacher et al, 2011; Murphy et al, 2014; Ray Chaudhuri et al, 2016; Taglialatela et al, 2017; Le et al, 2021). Thus, the BRCA2–RAD51 axis is central to the maintenance of genome integrity.

Deficiency of BRCA2 compromises RAD51 localization at DNA damage sites (Yuan et al, 1999; Moynahan et al, 2001; Xia et al, 2001; Godthelp et al, 2002; Tarsounas et al, 2003). Indeed, the recruitment of RAD51 to a damage site provides a significant opportunity for RAD51 regulation, subject to multiple auxiliary factors of RAD51 (Symington et al, 2014). Upon formation of DNA DSBs, DSB ends typically undergo resection, generating 3'-tailed single-stranded DNA (ssDNA). RAD51 eventually occupies exposed ssDNA, forming a nucleoprotein filament that samples double-strand DNA (dsDNA) for homology. Initially, the exposed ssDNA is occupied by RPA, the eukaryotic ssDNA–binding protein complex, which has a higher affinity for ssDNA than RAD51. Hence, the replacement of RPA with RAD51 requires a mediator complex, with BRCA2 playing a predominant role in higher eukaryotes, including humans (Sung et al, 2003).

Human BRCA2 is a large protein with a molecular weight of 384 kD, comprising multiple characteristic domains (Tavtigian et al, 1996; Le et al, 2021). The similarity between BRCA2 homologs from different species is largely limited to their C-terminus, which encompasses the helical domain and tandem oligonucleotide/oligosaccharide-binding domains (OB1, 2, and 3) (Lo et al, 2003). OB2 is inserted with the tower domain, consisting of a pair of long, antiparallel helices with a three-helix bundle at their end (Yang et al, 2002). ssDNA is located at the base of the tower domain, contacting OB2 and OB3 (Yang et al, 2002; Lo et al, 2003). The remaining part of BRCA2 is largely considered unstructured, featuring eight BRC repeats located around the protein's center (Bork et al, 1996; Lo et al, 2003). These BRC repeats serve as primary sites for RAD51 interaction (Wong et al, 1997; Davies et al, 2001; Pellegrini et al, 2002; Pellegrini & Venkitaraman, 2004). In addition, another RAD51 binding site that specifically binds to the RAD51 filament is located at the C-terminus (Mizuta et al, 1997; Sharan et al, 1997; Esashi et al, 2005).

BRCA2 forms a complex with DSS1, a small acidic protein serving as an obligate partner of BRCA2 (Marston et al, 1999). DSS1 binds to

Institute of Innovative Research, Tokyo Institute of Technology, Yokohama, Japan

Correspondence: htsubouchi@bio.titech.ac.jp; hiwasaki@bio.titech.ac.jp
Maierdan Palihati's present address is Department of Cancer Biology, The Cancer Institute of Japanese Foundation of Cancer Research, Koto-ku, Japan

BRCA2 by interacting with the helical and OB1 domains (Yang et al, 2002). Mutations in the BRCA2 that abrogate the interaction with Dss1 substantially reduce HR, demonstrating the importance of the interaction in HR function (Siaud et al, 2011). DSS1 stimulates BRCA2-mediated binding of RAD51 to RPA-occupied ssDNA (Zhao et al, 2015). Historically known as a component of the 26S proteasome complex (Paraskevopoulos et al, 2014), DSS1 is widely conserved across eukaryotes (Kragelund et al, 2016).

BRCA2 orthologs have not been found in popular yeast models like *Saccharomyces cerevisiae* and *Schizosaccharomyces pombe*. In these organisms, the role of a mediator is played by Rad52 instead. Rad52 forms an oligomeric ring structure and binds to both Rad51 and RPA in *S. cerevisiae*, promoting displacement of RPA from ssDNA to facilitate Rad51 loading onto ssDNA. Rad52 also has an activity that anneals complementary ssDNA strands (Shinohara et al, 1998; Sugiyama et al, 1998, 2006; Stasiak et al, 2000; Ranatunga et al, 2001). Interestingly, RAD52 coexists with BRCA2 in vertebrate species, where RAD52 seems nearly dispensable for HR (Rijkers et al, 1998; Liu & Heyer, 2011). The functional relationship between BRCA2 and RAD52 remains somewhat mysterious.

*Ustilago maydis*, a plant pathogen, is unique in that it is a fungal species carrying the BRCA2 homolog Brh2 (Kojic et al, 2002; Holloman et al, 2008; Holloman, 2011). *U. maydis* belongs to the Basidiomycota phylum, which is evolutionarily distant from Ascomycota to which *S. cerevisiae* and *S. pombe* belong. The two phyla diverged from a common fungal ancestor about 650 million years ago (Kumar et al, 2022). Basidiomycota, in general, has been explored to a lesser extent than Ascomycota. In this work, we directed our attention to *Naganishia liquefaciens*, a nonpathogenic Basidiomycota species (Abe et al, 2001). *Ustilago* (Ustilaginales) and *Naganishia* (Filobasidiales) diverged from a common ancestor about 450 million years ago (Han et al, 2020). We anticipate that studying multiple Basidiomycota species distantly related to each other will provide unique insights into basic biology, much like *S. cerevisiae* and *S. pombe*, two major Ascomycota models, have done.

The whole-genome sequence of *N. liquefaciens* has been drafted, and genetic tools enabling efficient gene targeting have been developed (Han et al, 2020; Palihati et al, 2021). In the course of examining genetic requirements for HR-mediated gene targeting, we found that Rad51 and Rad52 function redundantly in gene targeting when sufficient homology is provided (~1 kb) (Palihati et al, 2021). However, when homology is limited (80 bp), Rad52 is exclusively required. Interestingly, the absence of Rad52 hardly sensitizes cells to ionizing radiation and UV in *N. liquefaciens*. This is in sharp contrast to the *rad52* mutant of *S. cerevisiae* and *S. pombe*, where DNA damage repair becomes severely compromised. Similar subtle to no phenotypes in the absence of Rad52 are observed in vertebrate species and *U. maydis* (Rijkers et al, 1998; Yamaguchi-Iwai et al, 1998; Kojic et al, 2008).

In this study, we report the identification of the homologs of BRCA2 (hereafter called Brh2) and Dss1 in *N. liquefaciens*. BRCA2 homologs are also found in many Basidiomycota species but not in Ascomycota species like *S. cerevisiae* and *S. pombe*. The yeast two-hybrid assay demonstrated the interaction of Brh2 with both Rad51 and Dss1. Unlike vertebrate *BRCA2*, *BRH2* is not essential for viability. The loss of Brh2 caused pronounced sensitivity to UV and ionizing radiation, and their HR ability, as assayed by gene-targeting

efficiency, was compromised. These phenotypes were indistinguishable from those of the *rad51* mutant, and the *rad51 brh2* double mutant. Furthermore, the *brh2* defects in DNA damage and HR were exacerbated to a level similar to those of these two mutants when combined with the *rad52* mutation. These observations strongly support the argument that *Naganishia* Brh2 is the BRCA2 ortholog that functions as an indispensable auxiliary factor for Rad51.

## Results

### Identification of BRCA2 homolog in *N. liquefaciens*

We identified the coding sequence that encodes a BRCA2 homolog in the *N. liquefaciens* genome using the *U. maydis* Brh2 sequence as a query (DDBJ/EMBL/GenBank databases, BioProject accession number PRJDB10172). Its cDNA was sequenced, and the gene structure was determined (see the Materials and Methods section). The coding sequence encodes a protein consisting of 1,193 amino acids (accession #: GHJ86783, Fig 1A). This size is approximately one-third of the size of the human BRCA2 protein. The homology between *Naganishia* BRCA2 (Brh2) and human BRCA2 is limited to its C-terminal half, where major characteristic domains are clustered, including the helical domain and two oligonucleotide/oligosaccharide-binding domains (OB1 and OB2) (Fig S1). OB2

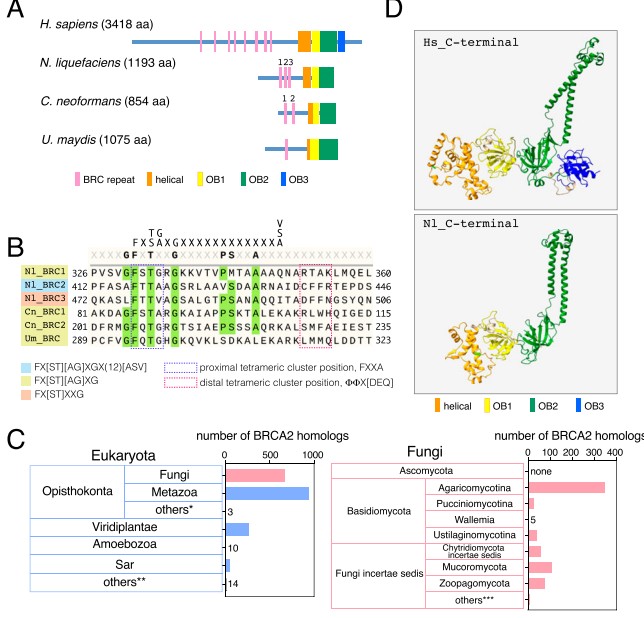

**Figure 1. BRCA2 homologs in *N. liquefaciens* and other Basidiomycota yeasts.**
**(A)** Schematic representation of *H. sapiens* (P51587) BRCA2 and its homologs in *N. liquefaciens* (GHJ86783), *C. neoformans* (UOH82896), and *U. maydis* (XP_011389639). **(B)** Alignment of potential BRC repeats from three Basidiomycota yeasts. Nl, *N. liquefaciens*; Cn, *C. neoformans*; Um, *U. maydis*. Numbers assigned to BRC repeats correspond to the numbering in (A). Φ, any hydrophobic amino acid. **(C)** Taxonomical distribution of BRCA2 homologs. Refer to Table S1 for more details. **(D)** Structural models of the BRCA2 C-terminal region of human (Hs) and *N. liquefaciens* (Nl) generated by AlphaFold2. Note that OB3 does not exist in *Naganishia* BRCA2.

distinctively contains an internal insertion called the tower domain. We observed that similarity between human BRCA2 and *Naganishia* Brh2 is limited along the sequence corresponding to the tower domain (Fig S1). On the other hand, the amino acids that interact with DSS1 in the mouse BRCA2 tend to be found within well-conserved regions (Fig S1) (Yang et al, 2002). The third OB fold, typically found in higher eukaryotes, is missing in *Naganishi* Brh2 and other fungal BRCA2 homologs.

Another signature motif called the BRC repeat is responsible for binding RAD51, the eukaryotic homologous recombinase. In human BRCA2, eight copies of relatively divergent BRC repeats are scattered around the middle of the protein. Initially, we applied the previously established BRC consensus, Fx[ST][AG]xGx(12) [ASV] to Brh2 (Lo et al, 2003) (Fig 1B, top). One sequence that matches the consensus was identified (designated as Nl_BRC2 in Fig 1B). However, another sequence that is similar to Nl_BRC2 but only satisfies Fx[ST]xxG was found downstream of Nl_BRC2, which was named Nl_BRC3 (Figs 1B and S2A). We then tested a mildly compromised consensus, Fx[ST][AG]xG, and Nl_BRC1 was identified.

Next, using *Naganishia* Brh2 as a query, we searched for BRCA2 homologs. Along with metazoa and viridiplantae (green plants), ~680 BRCA2 homologs were identified within the fungal kingdom (Fig 1C left and Table S1). Within fungi, more than 300 homologs are within Basidiomycota, which includes both *N. liquefaciens* and *U. maydis*, along with ~250 homologs found within the organisms categorized as fungi incertae sedis (fungi with uncertain placement) (Fig 1C right, Table S1). Remarkably, no homologs were identified in Ascomycota species.

*Cryptococcus neoformans* is a human pathogen that has recently emerged as a model organism for studying Basidiomycota biology. Both *C. neoformans* and *N. liquefaciens* are categorized under Agaricomycotina, and indeed, a BRCA2 homolog was found in *C. neoformans* as well (Fig 1A). The absence of OB3 is common to all three Basidiomycota BRCA2 homologs, likely a shared trait of fungal BRCA2 proteins (Fig S1). We applied the most stringent BRC repeat consensus (Fx[ST][AG]xGx(12)[ASV]) to *Cryptococcus* BRCA2 but found no matching sequence. With the consensus of intermediate stringency (Fx[ST][AG]xG), two sequences were identified (Fig 1B, Cn_BRC1, Cn_BRC2). In *Ustilago* Brh2, a single BRC repeat has been established that matches Fx[ST][AG]xGx(12)[ASV] (Fig 1B, Um_BRC) (Kojic et al, 2002). The BRC repeats form association with RAD51 through two tetrameric clusters of hydrophobic residues, FxxA and ΦΦx[DEQ], separated by 17 amino acids, with Φ representing hydrophobic amino acids (Rajendra & Venkitaraman, 2010). In the three Basidiomycota Brh2 proteins examined, reasonably conserved tetrameric clusters represented as FxT[GAV] were found at the proximal tetrameric cluster position of each BRC (Fig 1B). However, the amino acid sequences at the distal tetrameric position did not closely match the consensus, except for *Ustilago* BRC (Sutherland & Holloman, 2023). It is possible that the length of the spacer between the two tetrameric clusters is not conserved in *Naganishia* and *Cryptococcus* yeasts. The alignment of these BRC candidates highlights potentially conserved amino acids within Agaricomycotina, specifically GFxTxxGKxxxxPSxxA. Whether each serves as a bona fide BRC repeat, capable of interacting with Rad51, remains to be experimentally tested.

The similarity among BRCA2 homologs across various species is predominantly confined to the C-terminal region, where characteristic domains for DNA and Dss1 binding are clustered (Fig S1). Three-dimensional models of the C-terminal portion of both *Naganishia* Brh2 and human BRCA2 (spanning from the helical domain to OB2 for *Naganishia*, and from the helical domain to OB3 for human) were generated using AlphaFold2 (Fig 1D) (Jumper et al, 2021). The human C-terminal model largely aligns with the crystal structure of the mouse BRCA2 C-terminal region bound with Dss1 and ssDNA (Fig S2B) (Chen et al, 2008). In these models, the predicted structures show remarkable similarity, particularly in the tower domain, which is characterized by antiparallel helices extending from OB2, despite the absence of sequence similarity (Fig S1). The resemblance between these structures strongly implies the existence of similar biochemical properties.

Taken together, BRCA2 is widely conserved within Eukaryota, including fungi but excluding Ascomycota. The absence of OB3 appears to be a common trait among fungal BRCA2 proteins. The BRC repeat might exhibit greater divergence than previously thought, particularly among fungal species.

## Brh2 shows interaction with Dss1 and Rad51 on the yeast two-hybrid system

BRCA2 forms a complex with DSS1, an indispensable partner for BRCA2 function (Marston et al, 1999). A DSS1 homolog of *N. liquefaciens* was identified using *U. maydis* Dss1 as a query. Alignment of human DSS1 with Dss1 homologs from three Basidiomycota species (*N. liquefaciens*, *C. neoformans*, and *U. maydis*) revealed that two acidic regions are highly conserved across these species (Fig 2A).

We then examined the physical interaction between Rad51, Brh2, Dss1, and Rad52 using the yeast two-hybrid system. Brh2 fused to the Gal4 DNA-binding domain (DBD) showed interaction with Rad51 and Dss1 fused to the Gal4 transcription activation domain (GAD), and GAD-Brh2 also showed interaction with DBD–Rad51 and DBD–Dss1 (Fig 2B). These interactions play a central role in the function of BRCA2 and are consistent with its known properties in both higher eukaryotes and *Ustilago* BRCA2. Unexpectedly, we detected an interaction between Brh2 and Rad52. It would be particularly intriguing if there were a functional interplay between Brh2 and Rad52. Equally unexpected is the absence of an interaction between Rad51 and Rad52. This observation may be related to the seemingly limited involvement of Rad52 in HR in this organism (see the Discussion section) (Palihati et al, 2021). It is interesting to note that Dss1 did not show interaction with Rad52, given that DSS1 associates with RAD52 and enhances its activity in humans (Stefanovie et al, 2020).

To understand the function of Brh2 and Dss1 in *N. liquefaciens*, these genes were deleted, and the null mutants were subjected to further analysis. In the absence of Dss1, cells exhibited a marked reduction in growth (Fig 2C). Doubling time was ~120 min for the WT, *rad51*, and *brh2* mutants, whereas it was ~250 min for the *dss1* mutant. This extreme growth defect hindered us from further characterizing the *dss1* mutant. Dss1 is highly conserved, present in organisms that do not carry a BRCA2 homolog as well. It is a component of the lid subcomplex of the 26S proteasome, serving as an ubiquitin receptor (Paraskevopoulos et al, 2014).

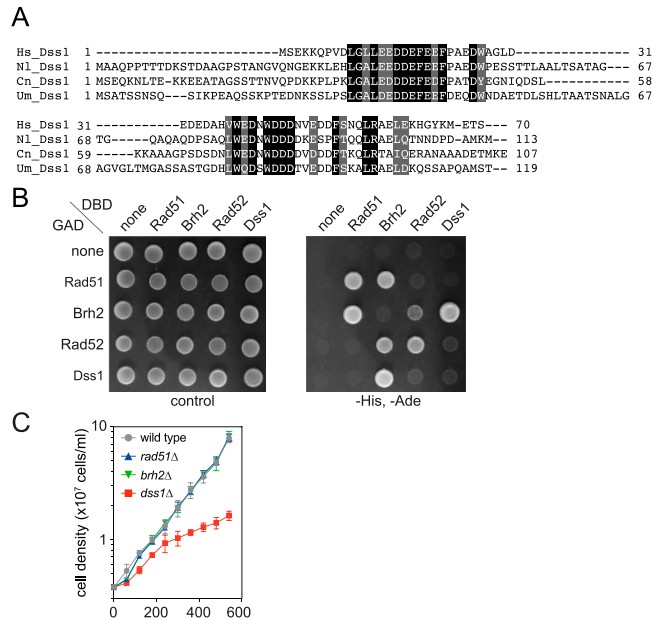

**Figure 2. Identification and characterization of Dss1 homologs in *N. liquefaciens*.**
**(A)** Alignment of Dss1 and its homologs. Hs, *H. sapiens*; Nl, *N. liquefaciens*; Cn, *C. neoformans*; Um, *U. maydis*. **(B)** Protein interactions assessed using the yeast two-hybrid assay. DBD, Gal4 DNA-binding domain; GAD, Gal4 transcription activation domain. "Control" indicates synthetic medium lacking tryptophan and leucin. "–HIS, –ADE" indicates a synthetic medium lacking histidine and adenine, and tryptophan and leucin. **(C)** Monitoring cell growth of various strains. Error bars, SD (n = 3).
Source data are available for this figure.

It is possible that Dss1 plays another BRCA2-unrelated function involving an essential protein in this organism. The absence of Brh2 or Rad51 did not result in a significant growth defect. This might implicate that the involvement of Rad51 and Brh2 in DNA replication is limited in this organism, in sharp contrast to vertebrate cells where Rad51 is essential for growth (Sonoda et al, 1998).

### *BRH2* and *RAD51* function in the same genetic pathway for DNA damage repair in *N. liquefaciens*

We then investigated the genetic relationship between *BRH2* and other HR genes previously characterized in *N. liquefaciens*, namely *RAD51* and *RAD52*. Single, double, and triple mutants of these genes were systematically tested for their sensitivity to UV or γ-ray. The absence of Brh2 sensitized cells to both UV and γ-rays to a similar extent as the absence of Rad51 alone or both Rad51 and Brh2. We previously demonstrated that the absence of Rad52 alone barely sensitizes cells to DNA damage, but when combined with the *rad51* mutation, cells display increased sensitivity to UV and γ-rays (Fig 3A and B left) (Palihati et al, 2021). The absence of Brh2 mirrored the effect caused by the absence of Rad51 in this context: the *brh2 rad52* double mutant exhibited practically the same sensitivity to both UV and γ-ray as the *rad51 rad52* double and the *rad51 brh2 rad52* triple mutants (Fig 3A and B right). Thus, these results strongly indicate

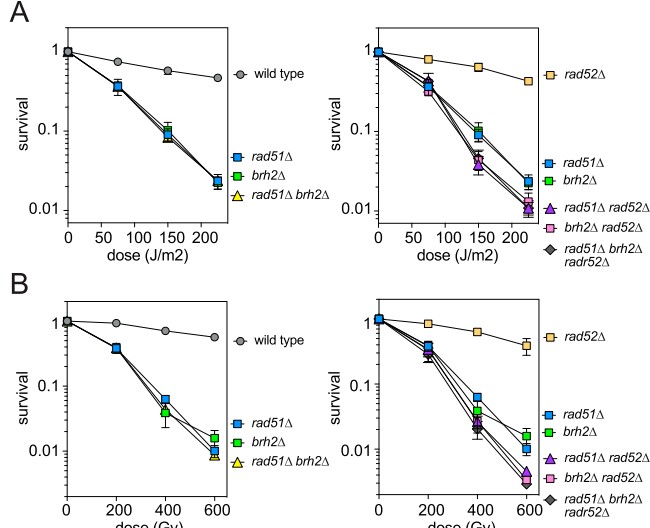

**Figure 3. *BRH2* and *RAD51* function in the same pathway in DNA damage repair.**
**(A)** UV and **(B)** γ-ray sensitivity results. Strains used: WT, MP1; *rad51Δ*, MP35; *brh2Δ*, MP31; *rad51Δ brh2Δ*, MP37; *rad52Δ*, MP33; *rad51Δ rad52Δ*, MP39; *brh2Δ rad52Δ*, MP41; *rad51Δ brh2Δ rad52Δ*, MP101. Error bars, SD (n = 3).
Source data are available for this figure.

that *BRH2* and *RAD51* operate in the same DNA damage repair pathway in *N. liquefaciens*.

### *BRH2* and *RAD51* function together in homologous recombination

We then investigated the involvement of *BRH2* in HR. Gene targeting is facilitated by HR-mediated mechanisms. We previously established that the length of homology determines the requirement for Rad51 and Rad52 in gene-targeting in *N. liquefaciens* (Fig 4A) (Palihati et al, 2021). We initially examined if the gene targeting of the *HIS3* gene with targeting DNA carrying 1 kb homologous arms is affected by the *brh2* mutation (Fig 4B left). The absence of Brh2 led to a mild reduction in gene-targeting efficiency, about 50% of that of the WT. This reduction is comparable with the level caused by the absence of Rad51 or both Rad51 and Brh2. With 1 kb of homologous arms, *RAD51* and *RAD52* function redundantly in gene targeting, rendering gene targeting completely defective in the *rad51 rad52* double mutant (Fig 4B left) (Palihati et al, 2021). Similarly, gene targeting was completely abolished in the *brh2 rad52* double mutant.

We previously demonstrated that the relatively low level of gene targeting in *N. liquefaciens* is attributed to the Ku70-dependent nonhomologous end-joining pathway (NHEJ) (Palihati et al, 2021). Indeed, gene-targeting efficiency increased to ~90% in the absence of Ku70, enhancing the sensitivity of gene-targeting measurements (Fig 4B right). The impact on gene targeting caused by the absence of either Rad51 or Brh2, or both, was limited if present. However, gene targeting was completely abrogated if either the *rad51* or *brh2* mutant was combined with *rad52*.

We then examined the gene-targeting conditions where targeting DNA has shorter homology arms (80 bp), a context where Rad52 is essential, whereas the Rad51 requirement is limited

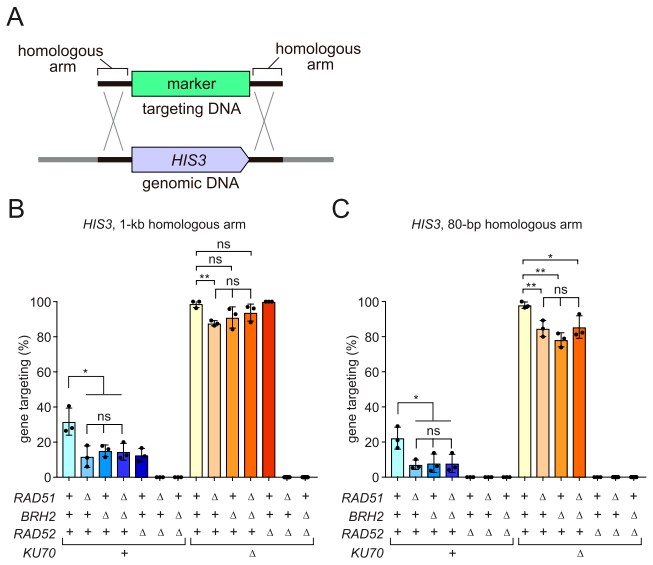

**Figure 4. Brh2 and Rad51 function in the same homologous recombination pathway.**
**(A)** Schematic drawing of the gene targeting system employed. **(B)** Gene-targeting efficiency with the targeting DNA with 1-kb homology arms. **(C)** The same as (B) except that the targeting DNA was with 80-bp homologous arms. Strains used are as follows: WT, MP1; *rad51Δ*, MP35; *brh2Δ*, MP31; *rad51Δ brh2Δ*, MP151; *rad52Δ*, MP33; *rad51Δ rad52Δ*, MP112; *brh2Δ rad52Δ*, MP139; *ku70Δ*, MP94; *rad51Δ ku70Δ*, MP108; *brh2Δ ku70Δ*, MP143; *rad51Δ brh2Δ ku70Δ*, MP131; *rad52Δ ku70Δ*, MP75; *rad51Δ rad52Δ ku70Δ*, MP113; *brh2Δ rad52Δ ku70Δ*, MP147. Error bars, SD. n = 3 in all the experiments except strains carrying *rad52Δ* (n = 9). Statistical significance was tested using unpaired two-tailed *t* test (ns, not significant; *$P < 0.05$; **$P < 0.01$; ***$P < 0.001$; ****$P < 0.0001$).
Source data are available for this figure.

([Palihati et al, 2021]). The absence of either or both Rad51 and Brh2 reduced gene targeting efficiency to ~30% of that of the WT ([Fig 4C] left). In the *ku70* mutant background, the absence of Rad51 or Brh2 marginally reduced gene targeting, by about 10%. In the absence of Rad52, regardless of the presence of Rad51 and Brh2, gene targeting was completely abolished ([Fig 4C] right).

Collectively, these genetic findings strongly support that Rad51 and Brh2 function together in HR-mediated gene targeting.

# Discussion

### *N. liquefaciens* Brh2 is the BRCA2 ortholog

In this study, we present the discovery of a BRCA2 homolog within the Basidiomycota yeast, *N. liquefaciens*. We propose that the *Naganishia* Brh2 homolog functions as the BRCA2 ortholog, substantiated by several lines of evidence. First, the C-terminal region, spanning from the helical domain to OB1 and OB2, exhibits significant similarity with BRCA2 orthologs from diverse species, including vertebrate BRCA2. Notably, the structural model of the C-terminus of *Naganishia* Brh2 shares remarkable resemblance with those of human and mouse BRCA2 proteins. This is particularly evident in the distinctive tower domain protruding from OB2, implying conserved functional attributes. Furthermore, we identified

the presence of the BRC repeat, a sequence motif characteristic of BRCA2, occurring in triplicate. Our investigation using the yeast two-hybrid assay demonstrated a robust interaction between Brh2 and Rad51, along with Dss1, an essential partner facilitating BRCA2 function.

Genetic analysis showcased parallels between the *brh2* mutant and both the *rad51* mutant and the *rad51 brh2* double mutant. Notably, all three mutants displayed a comparable phenotype, characterized by compromised DNA damage repair efficiency. Sensitivity to UV and γ-ray exposure was observed at similar levels, further exacerbated in the context of an additional *rad52* mutation. Similarly, our assessment of HR, as evaluated through gene targeting, unveiled a parallel reduction in efficiency within the *brh2*, *rad51*, and *rad51 brh2* double mutants. This convergence of phenotypes strongly suggests that Rad51's functionality in HR is contingent upon the presence of Brh2 in vivo. Consequently, Brh2 is deemed an indispensable auxiliary factor for Rad51 in *N. liquefaciens*. This study represents the first rigorous comparison of phenotypes resulting from the absence of Rad51 or BRCA2 with those stemming from the absence of both.

### DSB repair pathways in *N. liquefaciens*

The relationship between Rad52 and BRCA2 is intricate, as exemplified by the Ascomycota yeasts *S. cerevisiae* and *S. pombe*, both of which lack BRCA2. Instead, Rad52 assumes the role of a crucial auxiliary factor for Rad51. In these yeasts, Rad52 acts as a primary mediator, facilitating the replacement of ssDNA occupied by RPA with Rad51 ([Liu & Heyer, 2011]). Notably, the depletion of Rad52 in *N. liquefaciens*, analogous to findings in vertebrates and *U. maydis*, exerts negligible effects on DNA damage repair or HR ([Kojic et al, 2008]). Consequently, it is highly plausible that Rad52's role as an auxiliary factor for Rad51 is less prominent in this context.

Interestingly, the absence of Rad52 exacerbates DNA damage repair and HR deficiencies in the *rad51* or *brh2* mutant, indicating Rad52's involvement in HR-associated processes. This aligns with the extensively studied roles of Rad52 in Rad51-independent HR pathways, such as single-strand annealing and break-induced replication in *S. cerevisiae* ([Anand et al, 2013]; [Symington et al, 2014]). In organisms with BRCA2, Rad52 may primarily contribute to Rad51-independent functions. Notably, there was no interaction between Rad51 and Rad52 in the yeast two-hybrid assay ([Fig 2B]), which could suggest Rad52's specific involvement in a Rad51-independent function in *Naganishia*. In this context, it is worth mentioning that gene targeting with limited homology (80 bp) exclusively relies on Rad52 in *N. liquefaciens* ([Palihati et al, 2021]). It will be interesting to examine whether such a requirement for HR holds true in vertebrate species including humans.

It is intriguing to observe the interaction between Brh2 and Rad52 in the yeast two-hybrid assay, which suggests a potential interplay between these two proteins. Given the absence of interaction between Rad51 and Rad52 in the same assay, one possibility is that the interaction between Rad51 and Rad52 is mediated indirectly through Brh2 in this organism. Another possibility is the existence of an interplay between Rad52 and Brh2 in a Rad51-independent function. For instance, in vitro studies have shown that *U. maydis* Rad52 and Brh2 can both facilitate annealing of

ssDNA in vitro without the involvement of Rad51 (Mazloum et al, 2007). However, it is worth noting that the *brh2 rad51* double mutant essentially behaves like the *rad51* single mutant in terms of DNA damage sensitivity and gene-targeting assays, which argues against the presence of Rad51-independent roles for Brh2. In any case, comprehending the implication of the Brh2–Rad52 interaction holds the potential to shed light on new aspects of HR regulation in organisms carrying BRCA2.

We previously demonstrated that the NHEJ pathway predominates in DSB repair in *Naganishia* (Palihati et al, 2021). In *U. maydis*, Ku70, a major component of the NHEJ mechanism, is essential for growth, making it challenging to investigate the relationship between NHEJ and Brh2 (de Sena-Tomás et al, 2015). In contrast, the *ku70* deletion mutant is viable in *Naganishia*, allowing us to explore the genetic relationship between NHEJ and Brh2. In the *ku70* mutant background, where NHEJ is eliminated, gene targeting is predominantly mediated by HR. Under this condition, Brh2 and Rad52 redundantly facilitate gene targeting when sufficient homology (~1 kb) is available. However, Rad52 becomes indispensable when homology is limited to 80 bp. This relationship between Brh2 and Rad52 closely parallels the relationship between Rad51 and Rad52, establishing that Brh2 functions exclusively with Rad51 in the HR pathway when NHEJ is impaired.

### Conservation of BRCA2 across fungal species

The presence of BRCA2 homologs has been recognized in *U. maydis* (Kojic et al, 2002). We have extended this observation to the Basidiomycota yeast, *N. liquefaciens*. Our investigation has further identified ~680 BRCA2 homologs in various fungal species, with most of them falling under the phylum Basidiomycota. Notably, no BRCA2 homologs were identified within Ascomycota species. These observations align with a recent publication that reported the presence of BRCA2 homologs in all fungal phyla except Ascomycota (Sutherland & Holloman, 2023).

Among Basidiomycota, the Agaricomycotina class exhibits the highest prevalence of species possessing BRCA2 homologs. This class encompasses diverse organisms, including *N. liquefaciens* and *C. neoformans*, which is emerging as a model organism for studying Basidiomycota biology (Heitman et al, 2010). Together with *U. maydis* from the Ustilaginomycotina class, we compared three distinct Basidiomycota BRCA2 homologs. These homologs generally exhibit smaller sizes compared with their vertebrate counterparts, with about 1,000 amino acids, in contrast to human, chicken (*Gallus gallus*), and frog (*Xenopus laevis*) BRCA2, which have sizes of over 3,000 amino acids. Consistent with observations in other species, the sequence similarity of Basidiomycota BRCA2 homologs primarily resides in their C-terminal regions, encompassing the helical domain, OB1, and OB2. Notably, fungal BRCA2 proteins lack the OB3 domain commonly found in higher eukaryotes. Structural insights derived from the crystal structure of the mouse BRCA2 C-terminus, in complex with Dss1 and ssDNA, reveal that ssDNA resides at the base of the tower domain, making contact with OB2 and OB3 (Yang et al, 2002). Fungal BRCA2, only with OB1 and OB2, is expected to interact with DNA differently from BRCA2 with OB3.

The presence of putative BRC repeats has been observed in the Basidiomycota BRCA2 homologs (Fig 1B) (Sutherland & Holloman, 2023). However, these repeats are notably fewer in number compared with their human and vertebrate counterparts; for instance, only three BRC repeats are present in *N. liquefaciens*, whereas humans possess eight. Alignment of these Basidiomycota BRC repeat candidates reveals a potentially divergent BRC consensus sequence unique to Basidiomycota BRCA2. In the complex comprising human BRC repeat 4 (BRC4) and RAD51, the interaction is mediated by two tetrameric clusters of hydrophobic residues in BRC4, each occupying distinct hydrophobic pockets of RAD51 (Rajendra & Venkitaraman, 2010). The consensus sequences of these hydrophobic tetramers are FxxA and ΦΦx[DEQ], where Φ represents any hydrophobic amino acid. Both FxxA and FxxG are observed in fungal BRCA2 (Sutherland & Holloman, 2023). In *N. liquefaciens*, one of BRC candidate (Nl_BRC3) exhibits FxxV (Fig 1B). Moreover, none of the sequence at the position of the second hydrophobic tetramer perfectly matches the ΦΦx[DEQ] in *Naganishia* and *Cryptococcus* BRC candidates. These observations raise the possibility that BRC repeats of fungal BRCA2 are more divergent than previously thought, with the potential for more BRC repeats yet to be identified. It is important to note that the functional significance of these repeats and their conserved amino acids requires further biochemical validation, particularly through direct assessment of their interactions with Rad51.

In summary, the identification and thorough characterization of the Brh2 ortholog in *N. liquefaciens*, along with other fungal species, offer valuable insights into the fundamental mechanisms underpinning BRCA2's role and the intricate interplay between Rad51, Rad52, and Brh2. This exploration contributes to our comprehension of the general mechanisms governing genome stability and holds significant implications for cancer research.

# Materials and Methods

### Strains, growth conditions, and genetic methods

Strains and primers used in this study are outlined in Tables S2 and S3, respectively. *N. liquefaciens* was cultured following established methods (Palihati et al, 2021). Yeast cells were cultivated at 30°C in YPD medium (1% wt/vol yeast extract, 2% wt/vol Bacto Peptone, 2% wt/vol glucose) or a synthetic medium (0.67% yeast nitrogen base without amino acids and ammonium sulfate, 2% glucose, 0.5% ammonium sulfate, and 0.2% drop-out mix). Nourseothricin sulfate (GoldBio), G418 (Nacalai Tesque), and hygromycin B (InvivoGen) were added at 100, 100, and 25 μg/ml, respectively, when necessary. Standard genetic manipulations were conducted.

### Identification and analysis of BRCA2 homologs

We employed NCBI blastp to search for BRCA2 homologs, using the full length of *N. liquefaciens* Brh2 as the query with default settings, except for the following adjustments: database, nr_clustered; *E*-value, 0.005 (Sayers et al, 2022). Subsequently, the identified 2,381 clusters were filtered based on a query coverage of 25–100%. The resulting 1,608 clusters underwent taxonomical analysis, and the number of hits was compiled for kingdom, phylum, and class,

presented as "number of BRCA2 homologs." Similarity between BRCA2 homologs from various species was assessed and annotated using the NCBI constraint-based multiple alignment tool (Papadopoulos & Agarwala, 2007). Structure models for the C-terminus of human and *N. liquefaciens* BRCA2 were generated using AlphaFold2, and these models were annotated using UCSF ChimeraX 1.5 (Jumper et al, 2021; Pettersen et al, 2021).

### Transformation of *N. liquefaciens*

A fresh colony of *N. liquefaciens* was cultured in 5 ml of YPD medium at 30°C with shaking for ~15 h. Cells were then inoculated into 50 ml of fresh YPD ($0.2 \times 10^6$ cells/ml) and grown for 4–6 h until reaching the log phase ($1–2 \times 10^7$ cells/ml). Cell harvesting was achieved through centrifugation ($3,500g$, 5 min, 4°C). The pelleted cells underwent washing with ice-cold water and electroporation buffer (EB: 10 mM Tris–HCl [pH 7.5], 1 mM MgCl$_2$, and 270 mM sucrose). The cells were resuspended in 10 ml of EB containing 4 mM DTT. After incubation on ice for 15–30 min, cells were collected, washed with EB, and resuspended in 200 µl of EB. The cell suspension (45 µl) was mixed with 5 µl of DNA (3 µg) in a 0.2-cm electroporation cuvette (Bio-Rad) and used for electroporation (0.75 KV, 25 µF, and ∞ Ω; Bio-Rad Gene Pulser). Subsequently, electroporated cells were suspended in 1 ml of YPD and incubated at 30°C for 2 h before plating onto the appropriate selection medium. Typically, plates were incubated for 3 d at 30°C.

### Preparation of gene targeting fragments

Gene-targeting DNA with 80-bp and 1-kb homology arms was prepared as previously described (Palihati et al, 2021). In brief, PCR was conducted with a set of primers possessing 80-bp homology arms and pBS-Ptef1-NAT or pBS-Pact1-NEO as the template. For fragments with 1-kb homologous arms, DNA fragments flanking *HIS3* were first amplified by PCR using a set of primers (Pr-156-Pr-157, Pr-159-Pr-158) and *N. liquefaciens* N6 genomic DNA as the template. Subsequently, the 5′ and 3′ homologous arms were fused to a fragment carrying a drug-resistance marker through overlap extension PCR. PCR products were purified using the MonoFas DNA purification kit (GL Sciences).

### Strain construction

Gene replacement or editing was accomplished using the one-step replacement method or CRISPR/Cas9 technique. The latter employs pM101 that expresses Cas9 and the guide RNA scaffold (Palihati et al, 2021). For constructing *brh2::NAT*, targeting DNA was obtained through PCR with primers (Pr-401-Pr-402) and pBS-Ptef1:NAT as the template. In the case of *brh2::NEO*, primers (Pr-320-Pr-321) and pBS-Pact1:NEO were employed. Gene targeting was facilitated by pM101-gBRH2, which was created by cloning the annealed primers (Pr-342, Pr-343) at the BbsI site of pM101. Verification of successful gene targeting was performed through diagnostic PCR with primers (Pr-172-Pr-173). Similarly, for constructing *rad52::NAT*, gene-targeting DNA was obtained through PCR with primers (Pr-403-Pr-404) and pBS-Ptef1:NAT as the template. Gene targeting was conducted with pM101-gRAD52

(annealed Pr-361 and Pr-362 primers at the BbsI site of pM101), and correct targeting was confirmed using primers (Pr-174-Pr-175). The construction of *dss1::NAT* involved the preparation of two PCR fragments (Pr-140-Pr-112 and Pr-141-Pr-111), both using pBS-Ptef1:NAT as the template. The mixture of fragments was then used for gene targeting, and correct targeting was confirmed using primers (Pr-146-Pr-147). The creation of strain MP143 (*brh2::NAT ku70::NAT*) was achieved by transforming MP127 (*brh2::NEO ku70::HYG*) using a mixture of pM101-gNEO (annealed Pr-548 and Pr-549 primers at the BbsI site of pM101), the *brh2::NAT* donor fragment, pM101-gHYG (annealed Pr-611 and Pr-612 primers at the BbsI site of pM101) targeting the HYG cassette, and the *ku70::NAT* donor fragment. Nourseothricin-resistant clones sensitive to both hygromycin and G418 were selected, and gene replacement was confirmed through PCR (Pr-172-Pr-173 for *brh2::NAT*, Pr-504-Pr-505 for *ku70::NAT*). Strain MP147 (*brh2::NAT rad52::HYG ku70::NAT*) was constructed by transforming MP143 (*brh2::NAT ku70::NAT*) with a mixture of pM101-gRAD52 targeting *RAD52* with the *rad52::HYG* donor fragment. Hygromycin-resistant clones were selected, and gene replacement was confirmed through PCR (Pr-174-Pr-175 for *rad52::HYG*). Construction of *rad51::NAT*, *rad51::HYG*, *rad52::NAT*, *rad52::NEO*, *ku70::HYG* and *ku70::NEO* was described previously (Palihati et al, 2021).

### Cloning of Brh2 and Dss1 cDNAs

Total RNA was extracted from ~$1 \times 10^8$ cells at the late log phase using the NucleoSpin RNA kit (Macherey–Nagel). cDNA was synthesized using the ReverTra Ace-α- kit (Toyobo) as per the manufacturer's instructions. For this, 0.25 µg of total RNA was employed with the primers (*BRH2*, Pr-461; *DSS1*, Pr-568), or alternatively, 10 pmol of oligo (dT) 20 was used. The first strand synthesis was carried out at 42°C for 20 min. Subsequently, cDNA for *BRH2* or *DSS1* was PCR-amplified using primers listed in Table S2: *BRH2*, Pr-460-Pr-461; *DSS1*, Pr-567-Pr-568. The amplified DNA was sequenced, and the cDNA was cloned into pBluescript II SK(+) at the NdeI and BamHI sites, yielding pBS-*BRH2* and pBS-*DSS1*, respectively.

### Gene-targeting efficiency

Gene-targeting efficiency was determined as the ratio of the number of transformants displaying auxotrophic phenotypes to the total count of antibiotic-resistant transformants. Statistical analysis of differences in gene-targeting efficiencies between strains was performed using unpaired two-tailed *t* test (Graphpad Prism).

### DNA damage sensitivity assay

Cells were cultivated in 2 ml of YPD medium for ~24 h at 30°C. This culture was then used to inoculate 2 ml of fresh YPD medium ($0.2–0.5 \times 10^6$ cells/ml), and growth was continued until the cells reached the log phase. For spot tests, the cell suspension was adjusted to $1 \times 10^7$ cells/ml and serially diluted by 10fold. Next, 5 µl of each dilution was spotted onto YPD plates. For the clonogenic assay, cells prepared as described were spread onto YPD plates and subjected to UV or γ-ray irradiation at specified doses. Colony

counting was conducted after 3 d of incubation at 30°C, and the survival efficiency was calculated as the percentage of colonies on the irradiated plate relative to those on the unirradiated plate.

## Yeast two-hybrid assay

The Gal4-based matchmaker two-hybrid system 3 (Clontech) was used for the yeast two-hybrid analysis (Fields & Sternglanz, 1994). The indicated cDNAs were fused to the GAL4 activation domain (AD) in pGADT7 and the GAL4 DBD in pGBKT7. Specifically, pGADT7 and pGBKT7 were digested using NdeI and EcoRI. Rad51 cDNA was amplified using Pr-458 and Pr-459 primers, followed by digestion with NdeI and EcoRI, and subsequently inserted into pGADT7 and pGBKT7 vectors to form pM152 and pM154, respectively. Similarly, pGADT7 and pGBKT7 were digested using NdeI. Rad52 cDNA was amplified with Pr-550 and Pr-551, then cut with NdeI and cloned into pGADT7 and pGBKT7 to form pM200 and pM202, respectively. In addition, pGADT7 and pGBKT7 were also digested with NdeI and BamHI. Brh2 cDNA was amplified with Pr-460 and Pr-461 primers and Dss1 cDNA was amplified with Pr-567 and Pr-568 primers. These amplified fragments were cut with NdeI and BamHI and then inserted into pGADT7 and pGBKT7 vectors, resulting in the formation of pM153 and pM155, pM201 and pM203, respectively. These plasmids were introduced into the *S. cerevisiae* tester strain AH109. Transformants were selected on plates with synthetic medium lacking leucine and tryptophan, and reporter activation was assessed using a synthetic medium without leucine, tryptophan, histidine, and adenine.

# Supplementary Information

# Acknowledgements

We are grateful to the Biomaterials Analysis Division, Open Facility Center, Tokyo Institute of Technology for sequence analysis. We thank all members of the Iwasaki Laboratory for stimulating discussion. This work was supported in part by Grants-in-Aids for Scientific Research (A) (18H03985 and 22H00404 to H Iwasaki), for Scientific Research (B) (18H02371 and 23H02409 to H Tsubouchi) from the Japan Society for the Promotion of Science (JSPS), and by the Basic Research Grant from the Takeda Science Foundation.

## Author Contributions

M Palihati: conceptualization, data curation, investigation, methodology, and writing—original draft.
H Iwasaki: conceptualization, supervision, project administration, funding acquisition, and writing—review and editing.
H Tsubouchi: conceptualization, data curation, formal analysis, supervision, funding acquisition, validation, investigation, visualization, methodology, project administration, and writing—original draft, review, and editing.

## Conflict of Interest Statement

The authors declare that they have no conflict of interest.

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
