## [Reviewer comments · Life Science Alliance]

Life Science Alliance

Analysis of the indispensable Rad51 cofactor BRCA2 in *Naganishia liquefaciens*, a Basidiomycota yeast

Maierdan Palihati, Hiroshi Iwasaki, and Hideo Tsubouchi

DOI: <https://doi.org/10.26508/lsa.202302342>

Corresponding author(s): Hideo Tsubouchi, Tokyo Institute of Technology

Review Timeline:

Submission Date:	2023-08-26
Editorial Decision:	2023-10-03
Revision Received:	2023-11-06
Editorial Decision:	2023-11-16
Revision Received:	2023-11-16
Accepted:	2023-11-20

Transaction Report:

October 3, 2023

Re: Life Science Alliance manuscript #LSA-2023-02342

Dr. Hideo Tsubouchi
Tokyo Institute of Technology
Institute of Innovative Research
4259 Nagatsuta, Midori-ku
Yokohama, Kanagawa 226-8503
Japan

Dear Dr. Tsubouchi,

Thank you for submitting your manuscript entitled "The BRCA2 homolog of *Naganishia liquefaciens*, a Basidiomycota yeast, is an indispensable auxiliary factor for Rad51" to Life Science Alliance. The manuscript was assessed by expert reviewers, whose comments are appended to this letter. We invite you to submit a revised manuscript addressing the Reviewer comments.

Thank you for this interesting contribution to Life Science Alliance. We are looking forward to receiving your revised manuscript.

Sincerely,

B. MANUSCRIPT ORGANIZATION AND FORMATTING:

Reviewer #1 (Comments to the Authors (Required)):

This is a clearly written and thorough examination of the role of a BRCA2 ortholog in a little-studied fungus. It emphasizes that that *Saccharomyces* is exceptional in lacking this gene and shows clearly that BRCA2 in this organism plays an indispensable role in the Rad51 recombination pathway. The data are very clearly presented and convincing.

Reviewer #2 (Comments to the Authors (Required)):

This manuscript by Tsubouchi and colleagues describes identification of a BRCA2 homolog in the basidiomycete fungus *Naganishia liquifaciens* and a brief genetic characterization of the deletion mutant. The work is highly descriptive and includes only a minor amount of experimentation. It is performed well and the findings will be of interest to investigators in the field. The paper could be improved by some modifications outlined below.

The investigators should provide some rationale for why they are interested in developing this organism as an experimental system. How will this organism be of value in understanding the homologous recombination system beyond organisms already under study? Am I to understand that this organism was isolated from the Japan Trench - 6000 meters down in the Pacific Ocean? This is a salt water environment presumably with very low oxygen level, no light, and tremendously high pressure, yet the organism can grow in the laboratory on YPD medium? Does this organism not deserve some more comment as an extremophile?

Fig. 1-there is no mention of the two-module arrangement of hydrophobic tetrameric sequences in the BRC structure as elaborated by Venkitaraman and coworkers (Rajendra and Venkitaraman NAR 38, 82 [2010]) and no attention to aligning the consensus sequence of the second tetramer with the the putative BRC sequences. The putative Um_BRC1 of *Ustilago maydis* can be deleted without consequence (Kojic et al MCB 25, 2547 [2005]) so it might be a corrupted relic of a BRC element but is not functional and should not be discussed as a BRC repeat. Fungal BRC elements have been investigated and discussed previously (Sutherland and Holloman DNA Repair 127 2023).

Fig 1C and Table 1 are redundant. Eliminate one.

Fig 2B-Please show two hybrid panel with Rad52. Does it interact with Rad51? Dss1?

Fig. 2C-Devoting three panels to the growth rates of the mutants is excessive and redundant and of little value. The doubling times could be stated in the text in a single sentence.

The phenotype of the rad52 mutant is impressive and offers a chance to understand its role in HR in BRCA2 organisms. This is worth commenting on.

Another important finding that is not discussed is the viability of Ku70 mutants. This makes possible genetic studies on mutants doubly deficient in HR and NHEJ. This is a great experimental advantage.

The Methods section on strain construction indicates that Crispr/Cas9 was used, but no strains in the table of strains appears to have been constructed by that methodology.

The Title of the paper is not accurate and therefore inappropriate. Nowhere in the paper is it shown that Brh2 is an indispensable auxiliary factor for Rad51.

Reviewer #3 (Comments to the Authors (Required)):

By showing homology between BRCA2 and Brh2 in *N. Liquefaciens* the authors extend their search for BRCA2 homologs in *U.*

maydis, previously published. They show interactions between *N. Liquefaciens* Rad51/Brh1 and between Dss1/Brh1 by a two-hybrid assay. Unfortunately, the growth defect of *dss1D* strain did not allow to work with this strain further. Therefore, the authors compared the sensitivity of *brh2D*, *rad51D*, *rad52D* deficient strains and their combinations to DNA damaging agents. The sensitivity to UV and irradiation is interesting: *brh2D* strain is comparable to *rad51D*, indicating a major function of *brh2* probably in the same pathway of homologous recombination. With a simple and clever experiment describing homology search for long/short homologous regions, they show that *brh2D* and *rad51D* strains are affected in the same way during insertion of a fragment by homologous recombination.

This allows to conclude that Brh2 interacts with rad51 and probably acts in the same pathway.

A few comments on the manuscript:

The introduction is a bit confusing and difficult to follow. The main line is frequently not clear. There's a structure missing and the main line jumps between different subjects. This needs to be improved

Page 8 line 1. Figure 2B should be Figure 1B?

Page 8 line 6/7. 'BRCA2 homologs from ~680 BRCA2 homologs were identified'. This doesn't make sense.

Page16 Line 10-11: "Dss1 also associates with Rad52, stimulating its ssDNA annealing activity in *S. pombe* (Stefanovic et al., 2020)"

The cited paper does not say anything about Dss1 in *S. Pombe*. It refers to a paper below by saying that "RAD52 was very recently identified in the DSS1 interactome in *S. pombe* (62)"

62 Schenstrøm S.M., Rebula C.A., Tatham M.H., Hendus-Altenburger R., Jourdain I., Hay R.T., Kragelund B.B., Hartmann-Petersen R.. Expanded interactome of the intrinsically disordered protein *dss1*. *Cell Rep.* 2018; 25:862-870.

Please check properly the references in the manuscript.

The discussion on orthologs in different species is very disorganised making it difficult to follow and understand. It needs to be improved.

I am not sure about *rad51* paralogs in *L. naganishia*. Please add this point to the discussion.

Have you tested point mutants of Dss1 to overcome the problem of the growth initially encountered? Please add these experiments to the paper.

The authors didn't explain well why the interaction of Dss1 and Brh2 is important.

It might be interesting to identify point mutants of Dss1 to overcome the problem of the growth initially encountered? Such experiments could be added to the manuscript to extend its scope and potential significance.

Rad51D strain does not have a defect in cell growth in comparison to WT. Please explain this observation.

Comments to figures:

Fig1C: Where are placed Ascomycota discussed in the paper? Please add them to the diagram.

Fig1A and 1D: Why there are twice the same observations in the figure? They are redundant.

Fig1E: The structural similarity in alpha fold is not surprising regarding their conserved nature in a simple alignment.

We are very grateful to the reviewers for their constructive comments. Below, the original comments by the editor and reviewers are in blue italics, and our responses are in black. Any references to page/line numbers to indicate parts of the manuscript that have been modified are accurate for the revised, marked-up version of the manuscript. Major changes have been made in red throughout the manuscript.

Reviewer #1 (Comments to the Authors (Required)):

This is a clearly written and thorough examination of the role of a BRCA2 ortholog in a little-studied fungus. It emphasizes that that Saccharomyces is exceptional in lacking this gene and shows clearly that BRCA2 in this organism plays an indispensable role in the Rad51 recombination pathway. The data are very clearly presented and convincing.

The authors greatly appreciate this reviewer for the complimentary words.

Reviewer #2 (Comments to the Authors (Required)):

This manuscript by Tsubouchi and colleagues describes identification of a BRCA2 homolog in the basidiomycete fungus Naganishia liquifaciens and a brief genetic characterization of the deletion mutant. The work is highly descriptive and includes only a minor amount of experimentation. It is performed well and the findings will be of interest to investigators in the field. The paper could be improved by some modifications outlined below.

The investigators should provide some rationale for why they are interested in developing this organism as an experimental system. How will this organism be of value in understanding the homologous recombination system beyond organisms already under study? Am I to understand that this organism was isolated from the Japan Trench - 6000 meters down in the Pacific Ocean? This is a salt water environment presumably with very low oxygen level, no light, and tremendously high pressure, yet the organism can grow in the laboratory on YPD medium? Does this organism not deserve some more comment as an extremophile?

Basidiomycota has been explored to a lesser extent than Ascomycota in general. We proposed the use of *Naganishia liquifaciens* as a model organism for studying Basidiomycota biology (Palihati et al, 2021). Although we do not know whether or how *N. liquifaciens* actively proliferates in the deep sea environment, it has demonstrated robust growth under laboratory conditions using the same culture conditions and media for *S. cerevisiae*. Unlike *Cryptococcus neoformans*, *N. liquifaciens* is not a human pathogen. *N. liquifaciens* is very distantly related to *U.*

maydis while both falling within the Basidiomycota phylum (Han et al, 2020). We are hoping that this distinction will provide an opportunity to explore the diversity within Basidiomycota and potentially uncover insights that might not be apparent when studying a single species. The following text has been added to the introduction to clarify the biological significance of the study with this species (P. 6, line 18~):

Basidiomycota, in general, has been explored to a lesser extent than Ascomycota. In this work, we directed our attention to *Naganishia liquefaciens*, a non-pathogenic Basidiomycota species (Abe et al, 2001). *Ustilago* (Ustilaginales) and *Naganishia* (Filobasidiales) diverged from a common ancestor about 450 million years ago (Han et al, 2020). We anticipate that studying multiple Basidiomycota species distantly related to each other will provide unique insights into basic biology, much like *S. cerevisiae* and *S. pombe*, two major Ascomycota models, have done.

Fig. 1-there is no mention of the two-module arrangement of hydrophobic tetrameric sequences in the BRC structure as elaborated by Venkitaraman and coworkers (Rajendra and Venkitaraman NAR 38, 82 [2010]) and no attention to aligning the consensus sequence of the second tetramer with the the putative BRC sequences. The putative *Um_BRC1* of *Ustilago maydis* can be deleted without consequence (Kojic et al MCB 25, 2547 [2005]) so it might be a corrupted relic of a BRC element but is not functional and should not be discussed as a BRC repeat. Fungal BRC elements have been investigated and discussed previously (Sutherland and Holloman DNA Repair 127 2023).

I sincerely apologize for the oversight regarding the above-mentioned papers and greatly appreciate the comments.

The first BRC repeat of *Ustilago* (*Um_BRC1*) was removed from the alignment.

Regarding the two module arrangement of hydrophobic tetramers, we have reviewed the potential BRC repeats identified in *Naganishia* and *Cryptococcus* BRCA2 homologs, along with the unique BRC sequence of *Ustilago*. Two positions corresponding to the proximal and distal hydrophobic tetramers are annotated in the revised Figure 1B. Amino acid sequences at the position of the distal tetramer do not appear to match the proposed consensus sequence. This discrepancy may be attributed to the lack of conservation in the size of the spacer between the two tetramers.

The findings made in Sutherland and Holloman (DNA Repair 127 2023) were cited and discussed as follows (P. 20, line 8~):

The presence of putative BRC repeats has been observed in the Basidiomycota BRCA2 homologs (Fig. 1B) (Sutherland & Holloman, 2023). However, these repeats are notably fewer in number compared to their human and vertebrate counterparts; for instance, only three BRC repeats are present in *N. liquefaciens*, whereas humans possess eight. Alignment of these Basidiomycota BRC repeat candidates reveals a potentially divergent BRC consensus sequence unique to Basidiomycota BRCA2. In the complex comprising human BRC repeat 4 (BRC4) and Rad51, the interaction is mediated by two tetrameric clusters of hydrophobic residues in BRC4, each occupying distinct hydrophobic pockets of Rad51 (Rajendra & Venkitaraman, 2009). The consensus sequences of these hydrophobic tetramers are FxxA and $\Phi\Phi x[DEQ]$, where Φ represents any hydrophobic amino acid. Both FxxA and FxxG are observed in fungal BRCA2 (Sutherland & Holloman, 2023). In *N. liquefaciens*, one of BRC candidate (NI_BRC3) exhibits FxxV (Fig. 1B). Moreover, none of the sequence at the position of the second hydrophobic tetramer perfectly matches the $\Phi\Phi x[DEQ]$ in *Naganishia* and *Cryptococcus* BRC candidates. These observations raise the possibility that BRC repeats of fungal BRCA2 are more divergent than previously thought, with the potential for more BRC repeats yet to be identified. It is important to note that the functional significance of these repeats and their conserved amino acids requires further biochemical validation, particularly through direct assessment of their interactions with Rad51.

Fig 1C and Table 1 are redundant. Eliminate one.

Table 1 contains all the information, including the data omitted in Fig. 1C (the part annotated with asterisks). Since the majority of the content in Table 1 overlaps with Fig.1C, we have moved Table 1 to the Supplemental Materials as Table S1.

Fig 2B-Please show two hybrid panel with Rad52. Does it interact with Rad51? Dss1?

Naganishia Rad52 exhibited self-interaction, but it did not demonstrate interaction with Rad51. Interestingly it displayed interaction with Brh2. These new results have been presented in Figure 2B. The corresponding revision in the text reads (P. 12, line 18):

Unexpectedly, we detected an interaction between Brh2 and Rad52. It would be particularly intriguing if there were a functional interplay between Brh2 and Rad52. Equally unexpected is the absence of an interaction between Rad51 and Rad52. This observation may be related to the seemingly limited involvement of Rad52 in HR in this organism (see Discussion) (Palihati et al, 2021).

The discussion was also revised, which reads (P. 18, line 4~):

It is intriguing to observe the interaction between Brh2 and Rad52 in the yeast two-hybrid assay, which suggests a potential interplay between these two proteins. Given the absence of interaction between Rad51 and Rad52 in the same assay, one possibility is that the interaction between Rad51 and Rad52 is mediated indirectly through Brh2 in this organism. Another possibility is the existence of an interplay between Rad52 and Brh2 in a Rad51-independent function. For instance, *in vitro* studies have shown that *U. maydis* Rad52 and Brh2 can both facilitate annealing of ssDNA *in vitro* without the involvement of Rad51 (Mazloum et al, 2007). However, it is worth noting that the *brh2 rad51* double mutant essentially behaves like the *rad51* single mutant in terms of DNA damage sensitivity and gene-targeting assays, which argues against the presence of Rad51-independent roles for Brh2. In any case, comprehending the implication of the Brh2-Rad52 interaction holds the potential to shed light on new aspects of HR regulation in organisms carrying BRCA2.

Fig. 2C-Devoting three panels to the growth rates of the mutants is excessive and redundant and of little value. The doubling times could be stated in the text in a single sentence.

Only the graph showing growth curve was kept to avoid redundancy. The actual doubling-time has been just stated in the text.

The phenotype of the rad52 mutant is impressive and offers a chance to understand its role in HR in BRCA2 organisms. This is worth commenting on.

This point has been discussed in the original manuscript, and we have also reiterated it in this paper, along with our new yeast two-hybrid data (P. 17, line 14~):

Interestingly, the absence of Rad52 exacerbates DNA damage repair and HR deficiencies in the *rad51* or *brh2* mutant, indicating Rad52's involvement in HR-associated processes. This aligns with the extensively studied roles of Rad52 in Rad51-independent HR pathways, such as single-strand annealing and break-induced replication in *S. cerevisiae* (Anand et al, 2013; Symington et al, 2014). In organisms with BRCA2, Rad52 may primarily contribute to Rad51-independent functions. Notably, there was no interaction between Rad51 and Rad52 in the yeast two-hybrid assay, which could suggest Rad52's specific involvement in a Rad51-independent function in *Naganishia*. In this context, it is worth mentioning that gene targeting with limited homology (80 bp) exclusively relies on Rad52 in *N. liquefaciens* (Palihati et al, 2021). It will be interesting to examine whether such a

requirement for HR holds true in vertebrate species including humans.

Another important finding that is not discussed is the viability of Ku70 mutants. This makes possible genetic studies on mutants doubly deficient in HR and NHEJ. This is a great experimental advantage.

Thank you very much for your suggestion. The fact that the *ku70* deletion mutant is viable in this organism is indeed an advantage. This point has been included in the discussion (P. 18, line 17~):

We previously demonstrated that the NHEJ pathway predominates in DSB repair in *Naganishia* (Palihati et al, 2021). In *U. maydis*, Ku70, a major component of the NHEJ mechanism, is essential for growth, making it challenging to investigate the relationship between NHEJ and Brh2 (de Sena-Tomás et al, 2015). In contrast, the *ku70* deletion mutant is viable in *Naganishia*, allowing us to explore the genetic relationship between NHEJ and Brh2. In the *ku70* mutant background, where NHEJ is eliminated, gene targeting is predominantly mediated by HR. Under this condition, Brh2 and Rad52 redundantly facilitate gene targeting when sufficient homology (~1 kb) is available. However, Rad52 becomes indispensable when homology is limited to 80 bp. This relationship between Brh2 and Rad52 closely parallels the relationship between Rad51 and Rad52, establishing that Brh2 functions exclusively with Rad51 in the HR pathway when NHEJ is impaired.

The Methods section on strain construction indicates that Crispr/Cas9 was used, but no strains in the table of strains appears to have been constructed by that methodology.

The plasmid pM101, described in the section on strain construction, produces both Cas9 and gRNA, as previously detailed in our publication (Palihati et al, 2021). We have included a sentence describing what pM101 is to clarify our genome editing strategy (P. 24, line 7~).

The Title of the paper is not accurate and therefore inappropriate. Nowhere in the paper is it shown that Brh2 is an indispensable auxiliary factor for Rad51.

According to the reviewer's suggestion, we have changed the title to the following:

"BRCA2 homolog of *Naganishia liquefaciens*, a Basidiomycota yeast, is essential for Rad51 function"

Alternatively, if the criticism is toward "indispensability" of Brh2, the following title is also a possibility:

"BRCA2 homolog of *Naganishia liquefaciens*, a Basidiomycota yeast, is critical for Rad51 function"

Reviewer #3 (Comments to the Authors (Required)):

By showing homology between BRCA2 and Brh2 in N. Liquefaciens the authors extend their search for BRCA2 homologs in U. maydis, previously published. They show interactions between N. Liquefaciens Rad51/Brh1 and between Dss1/Brh1 by a two-hybrid assay. Unfortunately, the growth defect of dss1D strain did not allow to work with this strain further. Therefore, the authors compared the sensitivity of brh2D, rad51D, rad52D deficient strains and their combinations to DNA damaging agents. The sensitivity to UV and irradiation is interesting: brh2D strain is comparable to rad51D, indicating a major function of brh2 probably in the same pathway of homologous recombination. With a simple and clever experiment describing homology search for long/short homologous regions, they show that brh2D and rad51D strains are affected in the same way during insertion of a fragment by homologous recombination.

This allows to conclude that Brh2 interacts with rad51 and probably acts in the same pathway.

A few comments on the manuscript:

The introduction is a bit confusing and difficult to follow. The main line is frequently not clear. There's a structure missing and the main line jumps between different subjects. This needs to be improved

Apologies for any confusion. We have streamlined the introduction. It now begins with a general introduction to BRCA2, followed by the sections on the role of BRCA2 in HR, structural features of BRCA2, BRCA2 and Dss1, the relationship between BRCA2 and Rad52, Basidiomycota and fungal BRCA2, HR in *Naganishia*, and a summary of this work. I hope you find this arrangement more comprehensible.

Page 8 line 1. Figure 2B should be Figure 1B?

Thanks a lot for pointing this out. The corresponding part was corrected.

Page 8 line 6/7. 'BRCA2 homologs from ~680 BRCA2 homologs were identified'. This doesn't make sense.

The corresponding part was corrected as follows (P. 10, line 6~):

... ~680 BRCA2 homologs were identified within the fungal kingdom...

Page16 Line 10-11: "Dss1 also associates with Rad52, stimulating its ssDNA annealing activity in S. pombe (Stefanovie et al., 2020)"

The cited paper does not say anything about Dss1 in S. Pombe. It refers to a paper below by saying that "RAD52 was very recently identified in the DSS1 interactome in S. pombe (62)"

62 Schenstrøm S.M., Rebula C.A., Tatham M.H., Hendus-Altenburger R., Jourdain I., Hay R.T., Kragelund B.B., Hartmann-Petersen R.. Expanded interactome of the intrinsically disordered protein dss1. Cell Rep. 2018; 25:862-870.

Please check properly the references in the manuscript.

We apologize for any confusion. It was in humans, not in *S. pombe*. The introduction was restructured in the revised manuscript and the corresponding paper is no longer cited.

The discussion on orthologs in different species is very disorganised making it difficult to follow and understand. It needs to be improved.

We apologize for any confusion. To enhance readability, we have removed less relevant content and details. Information regarding *Drosophila* and *C. elegans* BRCA2 was removed. The revised paragraph reads as follows (P. 19, line 15~):

Among Basidiomycota, the Agaricomycotina class exhibits the highest prevalence of species possessing BRCA2 homologs. This class encompasses diverse organisms, including *N. liquefaciens* and *C. neoformans*, which is emerging as a model organism for studying Basidiomycota biology (Heitman et al, 2010). Together with *U. maydis* from the Ustilaginomycotina class, we compared three distinct Basidiomycota BRCA2 homologs. These homologs generally exhibit smaller sizes compared to their vertebrate counterparts, with about 1000 amino acids, in contrast to human, chicken (*Gallus gallus*), and frog (*Xenopus laevis*) BRCA2, which have sizes of over 3000 amino acids. Consistent with observations in other species, the sequence similarity of Basidiomycota BRCA2 homologs primarily resides in their C-terminal regions, encompassing the helical domain, OB1, and OB2. Notably, fungal BRCA2 proteins lack the OB3 domain commonly found in higher eukaryotes. Structural insights derived from the crystal structure of the mouse BRCA2 C-terminus, in complex with Dss1 and ssDNA, reveal that ssDNA resides at the base of the tower domain, making contact with OB2 and OB3 (Yang et al, 2002). Fungal BRCA2, only with OB1 and OB2, is expected to interact with DNA differently from BRCA2 with OB3.

I am not sure about rad51 paralogs in L. naganishia. Please add this point to the discussion.

This is a very interesting point. To date, we have identified at least three Rad51 paralog candidates based on sequence similarity, a contrast to *Ustilago*, where only Rec2 is known. However, it is evident that further characterization is required to confirm their status as genuine Rad51 paralogs and, if so, to determine which paralog corresponds to which. Given our limited knowledge of Rad51 paralogs at this stage, we believe it is premature to discuss (potential) Rad51 paralogs in the current manuscript.

Have you tested point mutants of Dss1 to overcome the problem of the growth initially encountered? Please add these experiments to the paper.

It might be interesting to identify point mutants of Dss1 to overcome the problem of the growth initially encountered? Such experiments could be added to the manuscript to extend its scope and potential significance.

Thank you for your suggestion. However, creating a clean *dss1* mutant that is specifically defective in its interaction with Brh2, while preserving other functions like those related to the proteasome, is challenging in the *Naganishia* experimental system, especially due to the absence of a plasmid system. Furthermore, we believe a comprehensible examination of the implications of the interaction between Brh2 and Dss1 is warranted in future study, as the Dss1 requirement can be readily bypassed by truncating the C-terminal DNA binding domain of Brh2 in *Ustilago* (Kojic et al, 2005). Therefore, we would prefer to proceed with the publication of this manuscript, focusing on the identification and basic characterization of Brh2 in *Naganishia*, without undertaking mutational analysis of Dss1 at this time.

The authors didn't explain well why the interaction of Dss1 and Brh2 is important.

Thanks a lot for pointing this out. The following sentences have been included in the introduction (P. 5, line 20~):

Mutations in the BRCA2 that abrogate the interaction with Dss1 substantially reduce HR, demonstrating the importance of the interaction in HR function (Siaud et al, 2011). Dss1 stimulates BRCA2-mediated binding of Rad51 to RPA-occupied ssDNA (Zhao et al, 2015).

Rad51D strain does not have a defect in cell growth in comparison to WT. Please explain this observation.

The following sentences have been included in the result (P. 13, line 10~):

The absence of Brh2 or Rad51 did not result in a significant growth defect. This might implicate that the involvement of Rad51 and Brh2 in DNA replication is limited in this organism, in sharp contrast to vertebrate cells, where Rad51 is essential for growth (Sonoda et al, 1998).

Comments to figures:

Fig1C: Where are placed Ascomycota discussed in the paper? Please add them to the diagram.

Thank you for pointing this out. Ascomycota has been included in Fig. 1C right.

Fig1A and 1D: Why there are twice the same observations in the figure? They are redundant.

I appreciate your suggestion. Fig. 1D was deleted.

Fig1E: The structural similarity in alpha fold is not surprising regarding their conserved nature in a simple alignment.

Within OB2, the amino acid sequences corresponding to the tower domain are not homologous (Fig. S1). We considered it important to demonstrate the structural conservation of this region. In the revised manuscript, we have placed greater emphasis on the fact that the insertion within OB2, corresponding to the tower domain in mouse BRCA2, is not conserved across eukaryotic species (P. 11, line 18~):

In these models, the predicted structures show remarkable similarity, particularly in the tower domain, which is characterized by antiparallel helices extending from OB2, despite the absence of sequence similarity (Fig. S1). The resemblance between these structures strongly implies the existence of similar biochemical properties.

November 16, 2023

RE: Life Science Alliance Manuscript #LSA-2023-02342R

Dr. Hideo Tsubouchi
Tokyo Institute of Technology
Institute of Innovative Research
4259 Nagatsuta, Midori-ku
Yokohama, Kanagawa 226-8503
Japan

Dear Dr. Tsubouchi,

Thank you for submitting your revised manuscript entitled "BRCA2 homolog of *Naganishia liquefaciens*, a Basidiomycota yeast, is essential for Rad51 function". We would be happy to publish your paper in Life Science Alliance pending final revisions necessary to meet our formatting guidelines.

- please address Reviewer 2's remaining comments, including suggestions for the title
- please upload your Tables in editable .doc or excel format
- please add the Twitter handle of your host institute/organization as well as your own or/and one of the authors in our system
- please add your main, supplementary figure, and table legends to the main manuscript text after the references section
- please consult our manuscript preparation guidelines <https://www.life-science-alliance.org/manuscript-prep> and make sure your manuscript sections are in the correct order
- there are callouts for Tables 1 and S4 in the manuscript text -- but these tables have not been provided -- please correct

A. FINAL FILES:

B. MANUSCRIPT ORGANIZATION AND FORMATTING:

Sincerely,

Reviewer #2 (Comments to the Authors (Required)):

The investigators have, for the most part, adequately addressed the points raised. The additional results provided for the two hybrid analysis are highly interesting. It is notable that there is interaction detected between Rad52 and Brh2. It is also interesting to note the lack of interaction between Dss1 and Rad52, in contrast to what was reported in the mammalian system. This surely also deserves comment in the text?

Sorry to raise this issue again but it is the opinion of this reviewer that the title is still inappropriate as it does not accurately portray what the paper is about.

The investigators have titled the work...

BRCA2 homolog of *Naganishia liquefaciens*, a Basidiomycota yeast, is (fill in the blank) for Rad51 function

previous title--- an indispensable auxiliary factor

current title--- essential

alternative suggestion--- critical

Maybe it is a language problem, but none of those titles accurately summarizes what the paper is about. Yes it is known from other systems that BRCA2 is essential, critical, indispensable for Rad51 function, but that is not what this paper shows. This paper describes Brh2 and its interaction with other homologous recombination genes in *Naganishia liquefaciens*. Why not say that?

Here are a some suggestions-

Analysis of the BRCA2 homolog of *Naganishia liquefaciens*, a Basidiomycota yeast

Elucidating the homologous recombination system of *Naganishia liquefaciens*, a Basidiomycota yeast

Analysis of the indispensable Rad51 cofactor, BRCA2, in *Naganishia liquefaciens*, a Basidiomycota yeast

We are again very grateful to the reviewer #2 for their constructive comments. Below, the original comments by the reviewer is in blue italics, and our responses are in black. Any references to page/line numbers to indicate parts of the manuscript that have been modified are accurate for the revised, marked-up version of the manuscript. Changes have been made in red.

The investigators have, for the most part, adequately addressed the points raised. The additional results provided for the two hybrid analysis are highly interesting. It is notable that there is interaction detected between Rad52 and Brh2. It is also interesting to note the lack of interaction between Dss1 and Rad52, in contrast to what was reported in the mammalian system. This surely also deserves comment in the text?

Thank you for pointing this out. We have commented on the observation with a proper reference, which reads (p. 12, line 22~):

It is interesting to note that Dss1 did not show interaction with Rad52, given that DSS1 associates with RAD52 and enhances its activity in humans (Stefanovie et al, 2020).

Sorry to raise this issue again but it is the opinion of this reviewer that the title is still inappropriate as it does not accurately portray what the paper is about.

The investigators have titled the work...

BRCA2 homolog of Naganishia liquefaciens, a Basidiomycota yeast, is (fill in the blank) for Rad51 function

previous title--- an indispensable auxiliary factor

current title--- essential

alternative suggestion--- critical

Maybe it is a language problem, but none of those titles accurately summarizes what the paper is about. Yes it is known from other systems that BRCA2 is essential, critical, indispensable for Rad51 function, but that is not what this paper shows. This paper describes Brh2 and its interaction with other homologous recombination genes in Naganishia liquefaciens. Why not say that?

Here are a some suggestions-

Analysis of the BRCA2 homolog of Naganishia liquefaciens, a Basidiomycota yeast

Elucidating the homologous recombination system of Naganishia liquefaciens, a Basidiomycota yeast

Analysis of the indispensable Rad51 cofactor, BRCA2, in Naganishia liquefaciens, a Basidiomycota yeast

Thanks a lot for your suggestion. The title has been changed to:

Analysis of the indispensable Rad51 cofactor BRCA2 in *Naganishia liquefaciens*, a Basidiomycota yeast

(100 characters including spaces)

November 20, 2023

RE: Life Science Alliance Manuscript #LSA-2023-02342RR

Dr. Hideo Tsubouchi
Tokyo Institute of Technology
Institute of Innovative Research
4259 Nagatsuta, Midori-ku
Yokohama, Kanagawa 226-8503
Japan

Dear Dr. Tsubouchi,

Thank you for submitting your Research Article entitled "Analysis of the indispensable Rad51 cofactor BRCA2 in *Naganishia liquefaciens*, a Basidiomycota yeast". It is a pleasure to let you know that your manuscript is now accepted for publication in Life Science Alliance. Congratulations on this interesting work.

DISTRIBUTION OF MATERIALS:

Again, congratulations on a very nice paper. I hope you found the review process to be constructive and are pleased with how the manuscript was handled editorially. We look forward to future exciting submissions from your lab.

Sincerely,
